# The Global Epidemiology of Bovine Leukemia Virus: Current Trends and Future Implications

**DOI:** 10.3390/ani14020297

**Published:** 2024-01-18

**Authors:** Guanxin Lv, Jianfa Wang, Shuai Lian, Hai Wang, Rui Wu

**Affiliations:** 1College of Animal Science and Veterinary Medicine, Heilongjiang Bayi Agricultural University, Daqing 163319, China; lgxarrron@foxmail.com (G.L.); wjflw@sina.com (J.W.); lianlianshuai@163.com (S.L.); 2Heilongjiang Provincial Key Laboratory of Prevention and Control of Bovine Diseases, Daqing 163319, China; 3China Key Laboratory of Bovine Disease Control in Northeast China, Ministry of Agriculture and Rural Affairs, Daqing 163319, China; 4College of Biology and Agriculture, Jiamusi University, Jiamusi 154007, China

**Keywords:** bovine leukemia virus (BLV), genomic structure, epidemiology, transmission, clinical impacts, diagnosis, control strategies

## Abstract

**Simple Summary:**

Bovine leukemia virus (BLV) is the causative agent of enzootic bovine leukosis (EBL), which is the most significant neoplastic disease in cattle. EBL is often overlooked in daily breeding processes due to the absence of obvious clinical symptoms. However, studies have revealed that EBL can severely impact the production performance of dairy cows, leading to a substantial economic burden on the cattle industry. In recent years, the global prevalence of EBL has been on the rise, and fragments of BLV nucleic acid have been detected in human breast cancer patients, raising public concerns. Due to the absence of an effective vaccine, controlling the disease is challenging. This review aims to provide a comprehensive overview of BLV, including its role in causing EBL, the genome of BLV, its current prevalence, transmission routes, clinical symptoms, detection methods, hazards, control strategies, and the current state of BLV research. The primary objective of this review is to offer breeders and researchers reliable veterinary knowledge on BLV and identify future research directions in this field.

**Abstract:**

Bovine leukemia virus (BLV) is a retrovirus that causes enzootic bovine leucosis (EBL), which is the most significant neoplastic disease in cattle. Although EBL has been successfully eradicated in most European countries, infections continue to rise in Argentina, Brazil, Canada, Japan, and the United States. BLV imposes a substantial economic burden on the cattle industry, particularly in dairy farming, as it leads to a decline in animal production performance and increases the risk of disease. Moreover, trade restrictions on diseased animals and products between countries and regions further exacerbate the problem. Recent studies have also identified fragments of BLV nucleic acid in human breast cancer tissues, raising concerns for public health. Due to the absence of an effective vaccine, controlling the disease is challenging. Therefore, it is crucial to accurately detect and diagnose BLV at an early stage to control its spread and minimize economic losses. This review provides a comprehensive examination of BLV, encompassing its genomic structure, epidemiology, modes of transmission, clinical symptoms, detection methods, hazards, and control strategies. The aim is to provide strategic information for future BLV research.

## 1. Introduction

Bovine leukemia virus (BLV) is a retrovirus known to cause enzootic bovine leucosis (EBL) [1,2,3,4]. EBL has been successfully eradicated in most European countries [5]. However, recent investigations have shown an increase in infection rates in several countries, including Argentina, Brazil, Canada, Japan, and the United States [6,7,8,9]. The majority of BLV-infected cattle remain asymptomatic, harboring the virus in a latent state of infection, and are thus classified as carriers. As the disease progresses, around one-third of cattle infected with BLV exhibit a non-malignant proliferation of B lymphocytes, a condition referred to as persistent lymphocytosis (PL). After a long period of latent infection lasting 3 to 8 years, less than 5% of these cattle will eventually develop malignant B lymphocyte lymphoma [10,11].

Previous research suggested that lymphosarcoma caused by BLV primarily affected the economics of cattle farming. However, recent research has shown that BLV infection affects the immune system of cattle, even during the latency period, leading to varying degrees of immune suppression and direct economic losses. These losses include reduced milk production, increased susceptibility to infectious diseases (such as mastitis, skin infections, and hoof diseases), and decreased reproductive efficiency [12,13,14,15]. In the United States, where the BLV infection rate exceeds 40%, the annual losses due to reduced milk production have surpassed USD 500 million. Additionally, certain countries and regions have implemented strict import and export regulations for animals infected with BLV, resulting in various indirect economic losses [5,13,15,16,17,18]. As a result, the World Organisation for Animal Health (WOAH) has classified Enzootic Bovine Leukosis (EBL) as a disease that significantly impacts international trade [19].

In addition to its impact on cattle herds, studies have shown that unsterilized or uncooked dairy and beef products contain proviral nucleic acid fragments of BLV, which raises concerns about public health and safety [20]. While it has not been confirmed that BLV can cause human infections, the positive detection rate of BLV in samples from breast cancer patients is significantly higher compared to healthy breast control groups [21,22,23,24,25,26]. Moreover, there is a correlation between the consumption of beef, milk, and dairy products and the incidence of breast cancer in different countries and regions. For example, countries with lower consumption rates like India, Japan, South Korea, and China also exhibit lower incidences of breast cancer. Conversely, countries with higher consumption rates such as the United States, the United Kingdom, Australia, and Germany have a higher incidence of breast cancer [27,28]. However, some studies have not detected proviral DNA of BLV in breast cancer tissue samples [29,30,31]. Therefore, further research is necessary to determine the impact of BLV exposure on human health, and government or industry associations should develop preventive measures to reduce the risk of BLV infection. Consequently, the objective of this review is to provide a comprehensive overview of current global BLV epidemiology and summarize recent advancements in BLV detection, diagnosis, and prevention methods in order to offer strategic information for future BLV research.

## 2. Genomic Composition of BLV

BLV is a single-stranded RNA-RT virus that exhibits a spherical or rod-shaped morphology, measuring approximately 60 to 125 nanometers in diameter. It is enveloped by a double-layered membrane structure and its nucleocapsid demonstrates icosahedral symmetry [32,33]. It contains a diploid single-stranded RNA genome. The viral genome of BLV consists of a diploid single-stranded RNA molecule spanning a total of 8714 nucleotides. This genome encompasses structural protein genes, enzyme coding genes, the pX region, and two identical, long, terminal repeat sequences positioned at both ends of the genome (Figure 1) [34,35].

Structural proteins and enzyme coding genes, such as *gag*, *pro*, *pol*, and *env*, are crucial in the virus lifecycle, infectivity, and the production of infectious viral particles [36,37,38]. The *gag* gene is highly conserved and responsible for encoding three major non-glycosylated proteins: p12 nucleocapsid protein (which binds with viral genomic RNA), p15 matrix protein (which binds with viral genomic RNA and interacts with the lipid bilayer of the viral membrane), and p24 nucleocapsid protein (which serves as the main target of the host immune response) [37]. The *pro* gene encodes the viral proteinase p14, which is involved in the post-translation maturation of the virus [36]. The *pol* gene encodes reverse transcriptase and integrase, which play a role in the reverse transcription process and integration of proviral BLV DNA into the host genome [39]. The *env* gene encodes two glycosylated proteins: envelope protein gp51 and transmembrane protein gp30, which facilitate virus–cell fusion through interaction with cell membrane receptors [40,41]. The pX region, located between the env gene and 3’LTR, encodes other auxiliary regulatory or non-structural genes, including regulatory proteins Tax and Rex, as well as auxiliary proteins R3 and G4 [42,43,44,45,46,47,48]. Regulatory proteins are important in virus transcription regulation, inducing the malignant transformation of tumors and virus infection output, while R3 and G4 auxiliary proteins help maintain a high viral load [49]. The *env* gene also shows genetic polymorphism, which is beneficial for phylogenetic studies and classifying BLV isolates [40]. The nucleotide sequence and amino acid composition of the env gene are valuable genomic markers for studying the distribution of BLV and identifying different genotypes based on geographical origin [50]. Researchers have identified 10 BLV genotypes through sequencing and phylogenetic analysis of partial and full-length gp51 *env* genes. Genotype 1 is the most prevalent and commonly found in the United States, Japan, and South Korea [51,52,53]. South America has genotypes 1, 2, 3, 4, 5, 6, and 9 (with genotype 9 being particularly prevalent in Bolivia). Russia and Eastern Europe commonly have genotypes 4, 7, and 8 [54]. Genotype 10 is prevalent in China, Vietnam, Thailand, and Myanmar [51,55]. Genotype 4 is found in various countries in North and South America, Africa, and Asia [56,57]. Genotype 6 has been found in South American countries such as Argentina, Brazil, Peru, Paraguay, and Bolivia, as well as in Asian countries such as the Philippines, Thailand, and India [56,58,59,60]. Additionally, genotype 1 has been reported in Egypt [61]. Monitoring the emergence of new viral mutations in veterinary medicine and animal husbandry is important since each new genotype may exhibit unique characteristics in its interaction with the host organism and leukemogenesis is influenced by specific genotypes (Figure 2).

Research indicates that 10 different microRNA (miRNA) can be transcribed under the control of RNA polymerase III in the genomic region spanning from the *env* gene to the R3 coding area [63]. RNA sequencing analysis has revealed that these miRNAs are highly expressed in primary sheep and cattle cells containing pre-leukemic and malignant B cells, accounting for 40% of the total cellular miRNA [64]. Additionally, elevated levels of BLV miRNA were found in the serum of infected animals, suggesting a potential paracrine role. These findings indicate the important role of BLV miRNAs in the retroviral cycle and/or disease progression, challenging the traditional belief that RNA viruses do not encode miRNAs. They also demonstrate that BLV miRNAs regulate various biological processes, including apoptosis, immunity, signal transduction, and tumorigenesis [65]. Notably, Blv-miR-B4-3p, which shares a seed sequence with the host miRNA miR-29a, is known for downregulating the tumor suppressor genes HMG-box transcription factor 1 (HBP1) and peroxidasin homologs (PXDN) in B-cell tumors [63,65,66]. In vitro studies have confirmed that blv-miR-B4-3p can directly downregulate these genes, although this has not been experimentally validated in cattle infected with BLV. Recently, Petersen et al. used RT-qPCR to analyze hbp1 and pxdn expression in cattle naturally infected with BLV and noted a significant downregulation of pxdn compared to uninfected animals [67]. However, a comprehensive analysis of targets for each mature BLV miRNA is still needed to fully understand the role of BLV in tumorigenesis.

## 3. Current Epidemiological Distribution of BLV

In the 19th century, research reports had already described clinical symptoms associated with EBL in cattle [68,69]. In 1871, Leisering published a study in Germany where he observed slightly yellow modules in the splenomegaly of cattle, along with an increase in white blood cells [68]. Bollinger further described and defined the clinical symptoms of EBL in 1874 [70]. In 1876, Siedamgrotzky and Hofmeister documented the first case of a bovine lymphocytic malignant tumor [71]. Subsequently, EBL started spreading globally, mainly due to the trade of dairy and beef cattle and their related products. In 1969, researchers successfully isolated the EBL pathogen from the lymphocytes of infected cattle and named it BLV [72].

Countries and regions with advanced economies and high consumption of beef, milk, and dairy products have shown greater concern for BLV. Since the 1960s, several countries and regions, including the United Kingdom, Denmark, the Netherlands, New Zealand, and Australia, have implemented successful BLV control programs, claiming to have eradicated the virus [73,74]. Their primary strategy has been to eliminate or isolate all cattle that test positive in the ELISA test [75,76]. On the other hand, countries like the United States, Canada, Argentina, and Japan have not adopted BLV control measures, resulting in a BLV prevalence exceeding 40% (Figure 3) [6,8,9]. Table 1 provides the BLV infection rates in different countries and regions [62].

The natural hosts for BLV infection are cattle (*Bos taurus*), zebu (*Bos indicus*), and buffalo (*Bubalus bubalis*). However, it has also been experimentally transmitted to sheep, goats, chickens, rabbits, and rats [109,110,111]. Studies have shown that genetic susceptibility plays a significant role in the prevalence of BLV in cattle populations [112]. For instance, cattle carrying the *BoLA-DRB3* gene have a significantly lower risk of BLV infection [113,114,115,116]. In dairy cattle populations, the frequency of BLV transmission is considerably higher compared to beef cattle populations [117]. A nationwide survey conducted in Japan from 2009 to 2011 reported seropositivity rates of 40.9% in dairy cattle and 28.7% in beef cattle [7,118]. Similarly, a study in China found infection rates of 49.1% in dairy cattle and 1.6% in beef cattle [96]. Furthermore, the seropositivity rate of BLV increases with the age of the cattle, as older age raises the risk of contact and infection [119,120].

## 4. Susceptibility and Transmission Pathways of BLV

BLV primarily resides in the peripheral blood lymphocytes of infected animals and can also be detected in various bodily fluids of BLV-positive cattle, including blood, milk, saliva, semen, and nasal secretions [121,122,123]. The main transmission routes of BLV are horizontal, which include (1) iatrogenic transmission, such as the reuse of medical devices contaminated with BLV (e.g., syringes, gloves, dehorning tools, castration tools, blood collection tools, and rectal examination instruments) during drug or vaccine injections [112,124]; (2) transmission through direct contact with infected cattle and their secretions, such as mucous membranes or the accidental ingestion of saliva, semen, urine, feces, and milk [124,125]; and (3) transmission through mosquito bites, including Stegomyia mosquitoes, Aedes vexans, and Aedes albopictus [126,127,128]. BLV can also be vertically transmitted from BLV-positive cows to their fetuses [129]. Additionally, BLV can be indirectly transmitted to fetuses through the ingestion of colostrum and milk containing free viral particles. This type of transmission typically occurs in dairy cows with a high proviral load (PVL) [130]. Studies have shown that cells in the milk of high-PVL dairy cows can carry BLV and transmit it indirectly through cell-to-cell contact [121]. The use of BLV-positive breeding bulls on farms can potentially result in the vertical transmission of BLV. While the risk of transmission through semen or embryos is considered to be negligible, natural mating with infected bulls can lead to transmission due to intense direct contact during mating [131,132]. Therefore, it is crucial to screen breeding bulls used for natural mating or artificial insemination to minimize the risk of BLV transmission [133].

## 5. Clinical Symptoms of EBL

EBL is divided into two types: fatal lymphoma-type EBL and sporadic bovine leukosis (SBL). The latter is non-contagious and primarily affects calves [11]. EBL progresses through three stages. In the first stage, approximately 70% of BLV-infected cattle test positive on serological tests but show no clinical symptoms or lymphocytosis. In the second stage, about 30% of BLV-infected cattle experience PL, which is characterized by non-malignant polyclonal expansion of CD5 + B cells. Most of these cattle carry high loads of BLV provirus. In the third stage, 5–10% of BLV-infected cattle eventually develop lymphosarcoma (LS) after a latent period of 3–8 years. This typically occurs in adult cattle and represents malignant proliferation [10,11,134,135,136].

The clinical symptoms of EBL include anorexia, digestive disorders, decreased milk production, bloating, abomasal displacement, diarrhea, constipation, superficial lymph node enlargement, lameness, paralysis, weight loss, and general weakness. In some cases, there may also be neurological dysfunctions [101,137]. Malignant lymphoma, caused by EBL, can invade various organs such as the uterus, mesentery, abomasum, spleen, lungs, kidneys, urinary tract, spine, scapula, and iliac lymph nodes, leading to disorders in the urinary, respiratory, and digestive systems [62,138]. Recent studies have revealed that infection with BLV reduces the diversity of microbial communities in the rumen and intestines of dairy cows, resulting in decreased energy conversion efficiency [139]. Furthermore, BLV infection impairs the function of macrophages and neutrophils, thereby suppressing the immune function of dairy cows. These findings provide insights into why BLV-positive dairy cows are more susceptible to other infectious diseases, experience decreased milk production, and have reduced reproductive efficiency [140,141].

## 6. Detection Methods of BLV

### 6.1. Serological Techniques Used for the Diagnosis of BLV

Researchers have developed various methods for detecting BLV, which can be broadly categorized into two types: serological tests based on BLV antibodies and polymerase chain reaction (PCR) tests based on the proviral DNA of BLV. Serological tests typically identify antibodies against the p24 capsid protein encoded by the *gag* gene and the gp51 membrane protein encoded by the *env* gene [142,143,144,145]. Common serological testing methods include agar gel immunodiffusion (AGID), passive hemagglutination assay (PHA), enzyme-linked immunosorbent assay (ELISA), and radioimmunoassay (RIA) [146,147,148,149,150]. These methods are suitable for detecting antibodies in bovine serum, milk, and the supernatants of BLV-infected cell cultures, as shown in Table 2 [142]. According to the WOAH, AGID and ELISA are the recommended tests for serological diagnosis of bovine leukemia virus (BLV) infection [151]. These tests have been validated using an international reference serum standard called E05, which establishes the minimum level of sensitivity required for the routine testing of serum and milk samples using AGID or ELISA.

However, each method has its own advantages and disadvantages. AGID is cost-effective and suitable for screening multiple samples but it has low sensitivity and is not suitable for milk samples. The efficiency of PHA is affected by pH values, temperature, and trypsin, which limits its reliability. RIA is suitable for immediate diagnosis after animal exposure but is not suitable for large-scale screening. On the other hand, ELISA is highly sensitive and easy to operate, making it suitable for serum and milk samples. However, it may produce false-negative results for early-stage serum samples and false-positive results for calves with maternal antibodies [99,142,144]. In summary, it is important to note that these antibody-based detection methods are not suitable for calves under six months old due to the potential for false-positive results caused by maternal antibodies.

### 6.2. Molecular Techniques Used for the Diagnosis of BLV

After BLV infection, genetic information is integrated into the host genome and remains transcriptionally silent for a long time to evade immune system recognition. In addition to conventional serological testing, various PCR techniques are widely used to directly detect the presence of proviral DNA in BLV-infected cattle with low, transient, or absent antibody titer. Various types of PCR are available for different purposes. Conventional single, semi-nested, and nested PCR tests are all valuable and sensitive tools for the early detection of BLV proviral DNA in blood, organs, or tumor samples. However, the semi-nested and nested tests offer significantly higher levels of sensitivity compared to single PCRs, as summarized in Table 3 [155,156,157,158,159,160]. Recently, a novel, blood-based PCR system was developed that can directly amplify DNA without the need for DNA separation and purification [161]. This method is highly specific and cost-effective, enabling the timely identification of BLV-infected cows. Takeshima developed a new qPCR-based method called BLVCoCoMo-qPCR for the quantitative detection of BLV. It is highly specific and sensitive, capable of detecting BLV strains in samples that test negative in nested PCR assays. This method has successfully detected various integrated BLV strains within the host genome from clinical cases of different geographic origins. With continuous technological improvements, BLVCoCoMo-qPCR can now measure the proviral load of both known and novel BLV variants, allowing for investigation into the correlation between BLV proviral load and disease progression [123,153,162,163,164]. For calves under six months of age that have not yet produced BLV antibodies, the PCR method is also applicable, but it requires specific laboratory equipment and instruments. 

Other diagnostic methods for BLV include the detection of BLV viral proteins through Western blotting (WB), syncytium formation assays in a co-culture, and indirect immunofluorescence (IF) for detecting BLV antigens [165,166,167,168,169,170].

**Table 3 animals-14-00297-t003:** Molecular techniques used for the diagnosis of BLV.

Test Assay	Sample Type	Advantages	Disadvantages	References
Conventional PCR (single, semi-nested, and nested PCR)	PBMC from blood, tumor sample, buffy coat, milk, somatic cells, semen, saliva, and nasal secretions.	♦ Direct, sensitive, rapid, and can detect recent infection of BLV even before the development of antibodies. ♦ Can be used for BLV detection during the early phase of infection or in the presence of colostrum antibodies.	♦ False negatives with low proviral load ♦ Cross-contamination ♦ Requires specific primers ♦ Requires equipment (PCR machine) ♦ Needs sequencing for confirmation	[152,158,171]
Quantitative (real-time)		♦ Direct, sensitive, rapid ♦ Differentiates EBL from SBL ♦ Detects BLV during the early phase of infection or in the presence of colostrum antibodies ♦ Quantitative measurement of proviral load	♦ Requires complicated sample preparation ♦ Requires positive controls ♦ Requires specific primers and probes ♦ Expensive and requires equipment (real-time PCR machine)	[123,153,162,163,164,169,172]
Direct, blood-based PCR	Blood	♦ Low cost ♦ Applied on the blood directly without DNA extraction or purification ♦ Low risk of contamination	♦ False negatives with low proviral load ♦ Low sensitivity	[161]

PCR testing can be used to quantify the PVL of animals infected with BLV using molecular techniques. PVL refers to the number of copies of the provirus per nanogram of host genomic DNA or in a certain number of infected cells. It is an important indicator for measuring the progression of retroviral diseases and can be used to assess the risk of transmission of human T-cell leukemia virus type 1 (HTLV-1) or the progression of human immunodeficiency virus (HIV) [173,174,175,176,177]. Similarly, the PVL of BLV can also serve as an indicator for assessing the risk of transmission and progression of EBL [156,178,179,180]. Previous research has classified a BLV proviral copy number exceeding 1000 copies/10 ng of DNA as a high proviral load (H-PVL) and values below this as a low proviral load (L-PVL) [163]. Studies have shown that cows with a H-PVL are more likely to transmit BLV to other cows, while cows with a L-PVL have a lower probability of transmission [181]. Other research has found a positive correlation between BLV PVL and increased lymphocyte counts in cows with a H-PVL compared to those with normal lymphocyte counts [182,183]. Additionally, BLV PVL is closely associated with factors such as the duration of cow herd use, milk production, milk quality, and the impairment of immune function [178,181,184].

PCR detection of BLV PVL has certain limitations. Studies have shown that there is significant variation in BLV PVL detection results when lymphocyte samples from cows that have tested positive for BLV by ELISA are tested in different laboratories [185]. This variation can be attributed to three main factors. Firstly, different laboratories use different PCR detection materials, such as various brands of Taq DNA polymerase and PCR equipment. Secondly, differences in the operating habits of personnel, even within the same laboratory, can result in up to a 10% variance in BLV PVL detection results for the same positive sample when different personnel use the same equipment and materials. Lastly, the use of different gene sequence primers by different laboratories can lead to a tenfold difference in BLV PVL detection results. Currently, there are no standardized procedures or sequence primers for BLV PVL detection. Therefore, to accurately measure BLV PVL, it is recommended that researchers compare and analyze BLV PVL detection values, the OD values of ELISA antibody titration, and lymphocyte counts. Existing studies indicate a positive correlation between BLV PVL and BLV antibody titers, as well as between lymphocyte counts and BLV PVL [113,186,187,188,189]. By establishing a relationship among these three factors, BLV PVL can be more accurately defined, facilitating scientific research and daily production practices.

## 7. Hazards of BLV

### 7.1. Effects on the Function of the Immune System of Dairy Cows

BLV significantly affects immune system function in dairy cows. Under antigen stimulation, B cells can transform into plasma cells, which are responsible for synthesizing and secreting antibodies, thus contributing to humoral immunity. BLV primarily targets IgM CD5+ B cells, leading to interference in antibody production and resulting in functional disorders [190]. Research indicates that BLV infection causes abnormalities in the synthesis and secretion levels of various antibodies in the body. For instance, PBMC freshly isolated from PL cattle shows increased expression of Igγ mRNA while exhibiting decreased levels of Igμ mRNA, indicating transcriptional disruption in antibody production [191]. Calves infected with BLV in experimental settings exhibit similar total serum IgG levels as uninfected calves, but they experience a temporary elevation followed by a decline in total serum IgM [192]. Moreover, BLV-infected cattle demonstrate compromised antibody production in response to specific antigens. Notably, PL cattle require twice the time compared to uninfected cattle to generate antigen-specific antibodies when exposed to synthetic antigens. Additionally, antibody production in PL cattle is inconsistent, lacking a stable antibody ratio, whereas uninfected cattle consistently maintain an IgM:IgG ratio of 1:10 [193].

BLV not only infects CD5+ B lymphocytes but also infects various subgroups of immune cells, including CD2+, CD3+, CD4+, CD8+, γ/δ T cells, monocytes, and neutrophils [15]. As a result, immune cell dysfunction occurs, manifesting as changes in cytokine production, surface receptor expression, cell proliferation, and apoptotic capabilities [15]. The production of cytokines by the immune system plays a crucial role in regulating the growth, differentiation, and immune responses of immune cells. In the case of BLV infection, Th1 cell dysfunction can be observed, leading to abnormalities in the secretion of various Th1 cell cytokines. Research has demonstrated that compared to BLV-infected cattle, the PBMC of AL cattle exhibits a significant decrease in the transcription levels of IL-4 and IFN-γ mRNA. Similarly, in PL cattle, the transcription levels of IL-2, IL-4, and IFN-γ mRNA in the PBMC are also significantly reduced [194]. Following ConA stimulation, the transcription level of IL-2 mRNA in the PBMC of PL cattle is significantly higher than that in uninfected cattle and asymptomatic (AL) cattle [195]. BLV infection also impacts IFN-γ secretion, with a significant reduction in the INF-γ mRNA transcription level observed in the PBMC of BLV-uninfected cattle compared to AL and PL cattle [194]. While there are no significant changes in the INF-γ mRNA transcription levels in AL and PL cattle after ConA stimulation, they still remain significantly higher than in uninfected cattle [195].

During the immune response to BLV, plasmacytoid dendritic cells (pDCs) are activated and secrete increased levels of IFN-γ, while conventional dendritic cells (cDCs) show decreased secretion of IFN-γ [196,197]. In BLV-infected cattle, there is a significant increase in IL-12 mRNA transcription levels compared to those in uninfected cattle, although this change is not prominent among BLV-infected cattle [197]. IL-12 has two subtypes: IL-12 (p40) and IL-12 (p70). The transcription level of IL-12 (p40) mRNA is significantly increased in the PBMC of AL cattle, while it is significantly decreased in PL cattle [198]. Furthermore, BLV infection activates the MoDc group, leading to a reduction in the expression levels of both IL-12 (p40) and IL-12 (p70) genes, which promotes the process of BLV infection [199,200]. BLV infection may also upregulate the expression of immunosuppressive molecules such as PD-L1, LAG-3, Tim-3, and CTLA-4 on the surface of PBMC through an unknown mechanism [201,202,203,204]. This results in a reduced response of Th1 cells and cytokine secretion, impairing the activity of T lymphocytes and diminishing antiviral functions. Helper T cells 2 (Th2) secrete cytokines such as IL-4, IL-5, IL-6, IL-10, and IL-13, which play a crucial role in assisting B cell activation. These cytokines promote the proliferation, differentiation, and generation of antibodies. BLV infection affects the transcription expression of certain Th2 cell factors, including IL-4, IL-6, and IL-10. Research has demonstrated that the level of IL-4 mRNA in the PBMC of BLV-infected cattle decreases, while no significant changes are observed in AL or PL cattle [195]. This suggests that BLV indirectly impacts the expression of IL-4 rather than directly regulating it. Additionally, studies have revealed that the expression level of IL-6 mRNA in the PBMC of PL cattle is higher than in other groups under the stimulation of ConA, lipopolysaccharide (LPS), and BLV-gp51 protein, indicating that BLV infection may lead to inflammation [193]. IL-10 possesses immunosuppressive properties and can inhibit the transcription of *Tax* and *pol* genes during BLV replication in PBMC. Furthermore, it has been observed that the secretion of IL-10 increases with the progression of BLV infection [199].

### 7.2. Effects on the Milk Production and Milk Quality of Dairy Cows

Milk production is a crucial measure of dairy cattle productivity and plays a significant role in the economic benefits of the dairy farming industry. However, there is still ongoing debate regarding the impact of BLV infection on milk yield in dairy cows. The National Animal Health Monitoring System Dairy Research Center of the United States Department of Agriculture conducted a study revealing that a higher rate of BLV-positive infection in cattle herds could potentially lead to a decrease in overall milk production, affecting high-yielding dairy herds to a greater extent [13,131]. However, when examining individual cows, no correlation was found between BLV infection and milk yield [13,131]. Other research demonstrated that milk production in BLV-positive cows is comparable to that of healthy cows, and, in some cases, it may even be higher [13,205]. Additionally, a four-year study on the milking performance of cows experimentally infected with BLV reported that artificial infection with BLV can actually enhance the milk yield of dairy cows [206]. Hence, further research is necessary to thoroughly understand the impact of BLV infection on milk production in dairy cows and its underlying mechanisms.

The demand for milk and dairy products has shifted from quantity to quality as society has progressed. To judge the quality of fresh milk, the components of milk, such as somatic cell count and bacterial count, are considered important criteria. Several studies have shown that cows infected with BLV have a significantly higher somatic cell count compared to that of uninfected cows [96,207]. The infection of BLV also affects the quality of dairy products. For instance, the milk of BLV-positive cows has significantly reduced levels of antimicrobial peptides and lactoferrin, which are related to viral load [178]. Furthermore, untreated fresh milk and raw beef can contain BLV nucleic acid sequences. Although there are currently insufficient data to establish a direct link between BLV and human breast cancer, BLV gene fragments have been detected in the tumor tissues of breast cancer patients [20,21,24,25,26]. Therefore, it is crucial to strictly test and control dairy products from BLV-positive cows to ensure food safety. Additionally, conducting further research on the impact of BLV on human health will contribute to understanding its potential risks and ensuring public health and safety.

## 8. Prevention and Control Strategies for BLV

Various strategies have been identified in previous studies for controlling and preventing BLV infection. These strategies include implementing strict management procedures, testing and culling or separating infected animals, genetic trait screening, and vaccination, summarized in Table 4 [16,182,183,208].

Numerous attempts have been made worldwide over the last few decades to reduce the impact of EBL on dairy herds by decreasing the prevalence of BLV-infected animals within the herd. Some positive effects have been observed in dairy herds when implementing strict management procedures. These practices aim to minimize the spread of BLV within the herd, and a thorough understanding of transmission modes is crucial for optimizing such practices. According to animal health law (regulation (EU) no. 2016/429) [76], the most important practices that should be followed are as follows. (1) Use only milk from cows that have tested BLV-negative or, alternatively, use a milk replacer to feed calves. If using milk from BLV-infected cows, it should be treated through freezing or heat treatment before feeding to the calves. (2) Consider using chemical dehorning or cautery methods to reduce the risk of infection. (3) When administering injections, always use disposable needles or sterilize reusable needles by boiling them between animals. (4) It is important to clean and disinfect ear tattoo implements thoroughly between animals to prevent the spread of infection. (5) Properly wash and disinfect stomach tubes and drenching guns after each use on an animal to avoid cross-contamination. (6) Use separate gloves for rectal exploration to prevent the transmission of diseases. (7) Ensure that all equipment used to assist with calving is thoroughly washed and disinfected to maintain a hygienic environment. (8) Separate calving paddocks should be designated for BLV-infected and uninfected cattle to minimize the risk of transmission. (9) It is advisable to remove calves from cows within 24 h of birth, but only after they have received an adequate intake of colostrum. (10) Implementing a fly control program can help reduce the presence of flies, which can transmit disease [209,210].

The eradication of a disease with a long incubation period can only be achieved through programs that target all infected animals, rather than just focusing on the visible outbreaks of the disease. Implementing stringent management tools in dairy herds can result in a reduction of within-herd prevalence, but it is important to note that it cannot completely eradicate the infection. Therefore, the most effective and sustainable approach to achieving freedom from BLV is by eliminating infected animals. Testing and culling is considered a primary approach, involving regular screening of cattle herds using ELISA or PCR techniques, followed by the culling of positive dairy cows. Some European countries have successfully eradicated BLV using this strategy [5,16]. Achieving freedom can only be possible when the rate of removing positive animals exceeds the annual infection rate. However, this task becomes increasingly challenging as the within-herd prevalence increases. Additionally, in countries where there is a high prevalence of BLV or weaker economies, it may not be feasible to implement testing and culling due to the high economic costs associated with it [12,183]. As a result, the success of this strategy depends on the support of national governments through economic compensation policies. Countries that do not have such policy support, such as the United States, Canada, Argentina, and Japan, have not implemented this strategy. In order to reduce the prevalence of the disease in a herd, a test and separate scheme can be implemented. This scheme aims to gradually decrease the prevalence to a level where it would be feasible to switch to a test and cull strategy. The success of this control strategy depends on the initial level of disease prevalence within the herd and the rate at which animals are culled [161,211]. Recent research has demonstrated that cows infected with BLV can be categorized based on BLV PVL, a crucial indicator for assessing transmission risk [172,212]. This categorization can serve as a standard for culling or separating infected cows. Cows with a high PVL generally exhibit higher peripheral blood lymphocyte counts, and routine blood tests can be employed to identify and cull these cows, thereby further reducing costs. In conclusion, both ‘test and cull’ and ‘test and separate’ strategies are viable for eradicating the disease, depending on the prevalence within a specific herd.

In the field of genetic breeding, selecting traits that are beneficial for production and farming, such as yield, growth speed, and reproductive efficiency, can significantly enhance efficiency. Similarly, achieving comparable results can be possible by choosing dairy cow breeds that are resistant to BLV infection. Immune responses and genetic resistance are influenced by the host’s major histocompatibility complex (MHC), specifically the bovine leukocyte antigen (BoLA) in cattle. Research studies have consistently shown that cattle carrying the BoLA class II *DRB3*0902* allele exhibit resistance to BLV or demonstrate significantly lower BLV proviral loads [213,214,215,216]. However, genetic breeding encounters certain challenges. It necessitates extensive research on a large scale to assess the effectiveness of marker genes in different breeds [217]. The genetic regulation of BLV is complex, making it challenging to consider specific alleles as absolute genetic markers [218,219]. Additionally, selecting based solely on genetic traits may have adverse effects on beneficial production traits [220]. Therefore, it is important to acknowledge that the strategies for BLV resistance selection may not always yield the expected benefits. 

In recent decades, scientists have dedicated considerable efforts to the development of vaccines against BLV. Initially, inactivated BLV vaccines were created using cell lines continuously infected with BLV, such as FLK, LK15, Bat2Cl1, and others [221,222,223,224,225,226]. These vaccines were successful in generating specific neutralizing antibodies in sheep and cattle, effectively protecting them against low-dose BLV infections. However, they were found to be ineffective against high-dose infections. As a result, researchers shifted their focus towards the design and development of BLV subunit vaccines. This approach was driven by the high conservation of the gp51 gene sequence, which is found in various BLV isolates and contains at least three neutralizing epitopes [224,225,226,227,228,229,230]. On the other hand, attempts to target the p24 protein in similar research were unsuccessful [227]. Consequently, synthetic peptide vaccines were developed to mimic the gp51, B-cell, and T-cell epitopes [231,232]. In sheep models, vaccines have been shown to induce significant humoral and Th1 responses, leading to a reduction in BLV replication in the short term [233,234]. However, subsequent experiments revealed that the synthetic peptide vaccine failed to sustain antibody induction or provide effective protection against BLV infection in most vaccinated sheep [233]. This poor performance could be attributed to the inadequate presentation of certain epitopes and stereochemical structures. As a result, researchers developed recombinant vaccinia virus (RVV) vaccines carrying BLV. Studies on RVV vaccines encoding gp51 alone revealed that RVV-gp51 could not induce a humoral response or protect sheep from BLV infection [235]. However, RVV-env vaccines encoding the complete *env* gene, including gp51 and gp30, successfully induced humoral immune responses in sheep models [236]. Despite this, these RVV-env vaccines were unable to induce specific neutralizing antibodies and showed limited efficacy in experiments with cattle [237,238]. On the other hand, DNA vaccines have the potential to induce long-lasting immunity. DNA vaccines containing the *env* and *tax* genes, controlled by cytomegalovirus or Srα promoters, demonstrated strong immune responses but were unable to prevent subsequent BLV infections [239,240]. Excitingly, an attenuated virus vaccine developed through gene deletion or mutation has been reported to effectively protect cattle from BLV infection [241]. However, the efficacy and biosafety of this vaccine require further validation. In summary, there are currently no safe and effective commercial vaccines available to control BLV infection in cattle.

## 9. Reducing the BLV PVL Contributes to BLV Prevention and Control

The life cycle of retroviruses consists of early and late stages. In the early stage, the virus invades cells and integrates with the host genome. In the late stage, viral RNA is expressed and viral particles are produced. Currently, most antiretroviral drugs are developed to target HIV. Highly active antiretroviral therapy (HAART) has significantly improved the prognosis of HIV patients [242]. Antiviral drugs used in HAART inhibit important steps such as viral entry, reverse transcription, integration, and processing [243]. These drugs specifically target receptors, reverse transcriptase, integrase, and protease. Therefore, developing compounds that target these steps could be a promising treatment for BLV. However, previous studies have shown that reverse transcriptase inhibitors used for HIV have limited effects on BLV [35]. Recently, a study discovered a natural compound called Violacein E that can inhibit BLV replication in experiments [244]. However, these studies are still in the early stages and further extensive research is needed before practical application of these compounds can be considered. Additionally, some natural compounds are expensive and have complex extraction processes, and their effectiveness has not met expectations. Overall, due to the high incidence of BLV, the search for anti-BLV compounds remains a primary focus in treatment. However, there is still a need to develop commercially available drugs that can effectively reduce the BLV PVL.

During BLV infection, the body undergoes varying degrees of immune suppression, which can be attributed to the Th1 cell immune response. Investigating the mechanisms behind immune function suppression during BLV infection is crucial for identifying novel treatment strategies. Studies have revealed that the expression of immune suppressive molecules such as PD-L1, LAG-3, TIM-3, and CTLA-4 increases during BLV infection [201,202,203,204]. These molecules have been shown to hinder BLV-specific Th1 responses and contribute to disease progression. Research conducted by Okagawa T and Nishimori A suggests that blocking the PD-1/PD-L1 pathway with antibodies significantly reduces the BLV PVL, indicating that antibody blockade is an effective approach for treating BLV [245]. However, the production of such antibodies is expensive and not suitable for large-scale application in BLV control and treatment. In addition to this, BLV infection disrupts the functions of Th1 and Th2 cells, and cytokines demonstrate complex characteristics such as pleiotropy, antagonism, and synergy. Abnormal cytokine secretion during BLV infection is another important factor contributing to immune function suppression. Therefore, drugs that modulate the functions of Th1 and Th2 cells, restore cytokine secretion patterns, and enhance antiviral functions could potentially help reduce BLV PVL [35].

## 10. Conclusions

The prevalence of EBL poses a significant threat to the livestock industry and has resulted in substantial economic losses. Currently, the most effective strategy for controlling EBL involves identifying and eliminating infected animals, as well as implementing trade protection measures to prevent the entry of these animals. Most European countries have successfully employed this method to control EBL, despite its high cost. Furthermore, EBL control can also be achieved through genetic selection, management interventions, and the development of effective vaccines. While EBL cannot be completely cured, medication can be used to reduce the BLV PVL, thereby reducing the risk of transmission.

## 11. Future Prospective Studies

Vaccines are widely recognized as the most effective means of reducing disease transmission. However, developing a vaccine for retroviruses is particularly challenging. Despite decades of efforts, effective vaccines for retroviruses like HIV and HTLV-1 have not yet been developed. One of the reasons for this is the lack of suitable animal models and the long incubation period from infection to onset, which makes it difficult to accurately understand the disease progression. Additionally, BLV, a retrovirus that causes significant economic losses, has received less research attention compared to human or other bovine diseases. Instead, BLV has been primarily studied as a research model for HTLV-1 and other retroviruses. Drawing from the lessons learned from previous vaccine failures, it is recommended to develop attenuated vaccines for BLV by deleting single or multiple genes associated with viral immunosuppression or virulence. Another approach could be to learn from the development methods of COVID-19 mRNA vaccines. In addition to vaccine development, it is crucial to focus on effectively reducing BLV PVL as numerous studies have confirmed their correlation with the risk of BLV transmission. High-throughput in vitro cell assays can be used to screen natural compounds that effectively reduce BLV PVL. By reducing PVL and implementing detection and elimination strategies, the global spread of BLV can be controlled. Furthermore, the integration of the latest deep sequencing technology can facilitate in-depth research on the BLV transcriptome, metabolome, and proteome. A comprehensive understanding of the interaction between BLV and the host will provide valuable knowledge and insights for future research on BLV.

## Figures and Tables

**Figure 1 animals-14-00297-f001:**
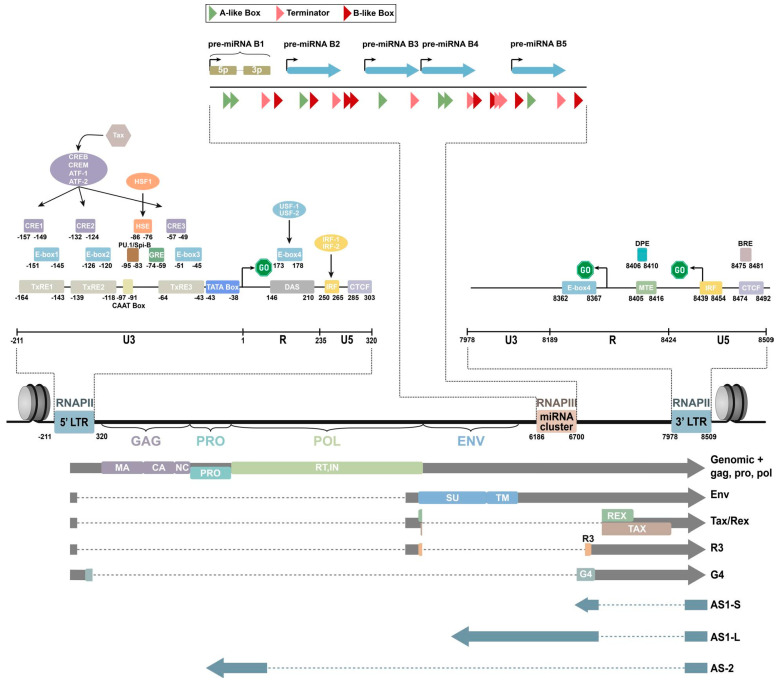
Schematic representation of the BLV genome [35].

**Figure 2 animals-14-00297-f002:**
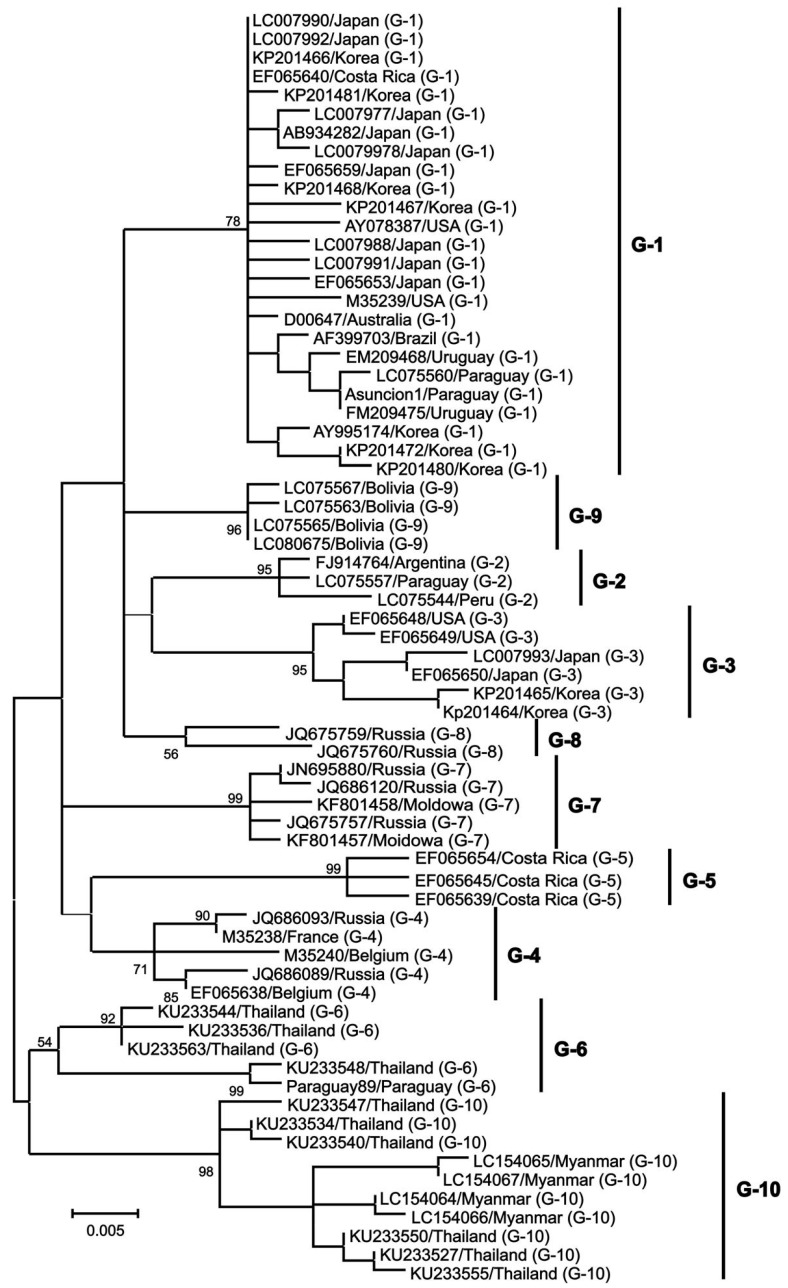
A maximum likelihood phylogenetic tree was constructed using partial BLV *env* sequences from various geographical locations worldwide, from [62].

**Figure 3 animals-14-00297-f003:**
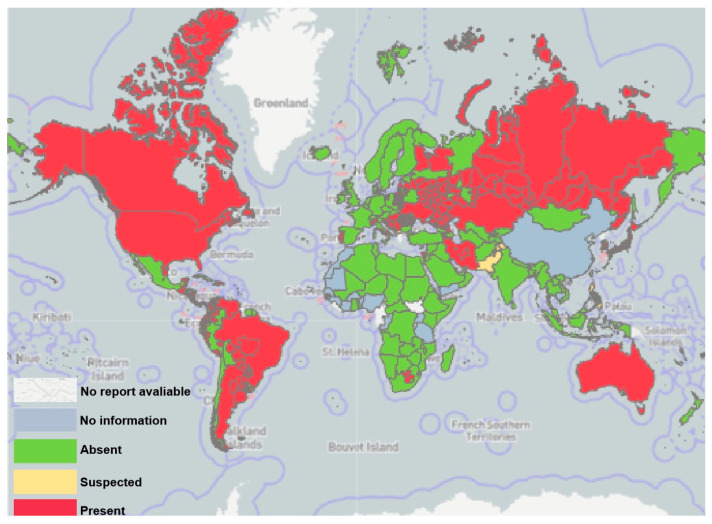
World distribution map of EBL based on the last 5 years (2019–2023), from the WOAH database WAHIS [77].

**Table 1 animals-14-00297-t001:** Prevalence of EBL worldwide, updated and modified from [62].

Status	Continent	Countries	Year	References
BLV-free	Europe	Belgium	2016 2021	[5,78,79]
		Czech Republic		
		Denmark		
		Germany		
		Estonia		
		Ireland		
		Spain		
		France		
		Italy		
		Cyprus		
		Latvia		
		Lithuania		
		Luxembourg		
		The Netherlands		
		Austria		
		Poland		
		Portugal		
		Romania		
		Slovenia		
		Finland		
		Sweden		
		United Kingdom (Northern Ireland)		
	Oceania	Australia	2013	
		New Zealand	2008	
		Tunisia	2005	
	Asia	Kyrgyzstan	2008	
		Kazakhstan	2007	
BLV existing in countries with unknown prevalence	Europe	Belarus	Present	
		Bulgaria		
		Croatia		
		Greece		
		Ukraine		
BLV existing in countries with variable prevalence	North America	Canada 78% at herd level 88.39% at herd level 89.30% at herd level	1998–2003 2016 2018	[80,81]
		Mexico Dairy 36.1%, beef 4% Dairy 50.6%	1983 2023	[82,83]
		USA Dairy 83.9%, beef 39%	2007	[84]
	Europe	Russia 22.1% to 25.4%	2012	[85]
	South America	Argentina (Buenos Aires) 90.9% at herd level Argentina (multiple regions) 84% at herd level 90.16%	1998–1999 2007 2011	[56,86,87]
		Bolivia (multiple regions) 30.7% at individual level	2008	[56]
		Brazil 17.1% at herd level 60.8% at herd level 23.4% at herd level	1980–1989 1992–1995 2021	[88,89,90]
		Chile (southern regions) 27.9% at individual level	2009	[56]
		Colombia 62% at individual level	2020	[52]
		Ecuador 5.6% at individual level	2012	[91]
		Paraguay (Asuncion) 54.7% at individual level	2008	[56]
		Peru (multiple regions) 31% at individual level, 42.3% at individual level	1983 2008	[56,92]
		Venezuela 33.3% at individual level 60.83% at individual level	1978 2011	[93,94]
		Uruguay 10.4% at individual level	2013	[95]
	Asia	China Dairy 49.1%, beef 1.6%	2013–2014	[96]
		Cambodia Draught cattle 5.3%	2000	[97]
		Iran (nationwide) 22.1% to 25.4%	2012–2014	[98]
		Myanmar 9.1% at individual level	2016	[56]
		Japan (nationwide) Dairy 49.1%, beef 1.6% 79.1% at dairy herd level 73.3% at individual level	2009–2011 2007 2012–2014	[6,99]
		Mongolia Dairy 3.9%	2014	[100]
		The Philippines 4.8% to 9.7%	2010–2012	[101]
		Pakistan Dairy 20%	2019	[102]
		Taiwan 81.8% at animal level, 99.1% at herd level	2019	[103]
		Thailand 58.7% at individual level	2013–2014	[60]
	Middle East	Israel 5% at individual level	2005	[104]
		Saudi Arabia Dairy 20.2%	1990	[105]
		Turkey Dairy 48.3%	2005	[106]
		Iraq Dairy 7%	2015	[107]
		Egypt Dairy 17.7%	2020	[108]

**Table 2 animals-14-00297-t002:** Serological techniques used for diagnosis of BLV.

Test Assay	Sample Type	Target	Advantages	Disadvantages	References
AGID	Serum	Antibodies (p24, gp51)	Specific, simple, rapid, and low screening cost	Less sensitive, inconclusive, and fails to evaluate disease states	[137,147,148,149,152]
ELISA	Serum, milk, bulk milk	Antibodies (p24, gp51)	Sensitive, specific, large-scale screening, and rapid	False negatives (particularly in cattle during the early stages of infection) False positives (maternally derived antibodies) Cannot evaluate disease states of infected cattle	[142,143,148,149,153]
RIA	Serum	Antibodies (p24)	Sensitive in detecting BLV at an early stage of infection	Cannot be used for mass screening	[154]
PHA	Virus particle	BLV glycoprotein	Sensitive, specific, rapid, and low screening cost	Affected by pH and temperature Hemagglutination activity reduced by trypsin and neuraminidase	[150]

**Table 4 animals-14-00297-t004:** Control tools for EBL, modified from [76].

Goal/Method	Tool/Components	Applicability
**Eradication** (elimination of infected animals)	**Test and slaughter** ♦ Regular testing and promptly culling infected animals ♦ Culling of the offspring of infected animals ♦ Safe management practices should be followed to prevent the spread of the virus among animals	**Small heard, with low herd BLV prevalence, supported for restocking** ♦ Low or moderate within-herd prevalence ♦ Freedom from the infection can be achieved if the rate of removal of positive animals exceeds the annual incidence rate of infection ♦ Support of national governments through economic compensation policies
	**Test and separate** ♦ Crucial to physically separate infected cattle from uninfected cattle ♦ Gradual elimination of infected animals can be achieved by increasing the frequency of culling in the infected group. ♦ Regular testing should be conducted on the seronegative group, and any positive animals should be promptly separated or eliminated ♦ In the final stage of the eradication program, a ‘test and slaughter’ strategy is implemented ♦ Safe management practices should be followed to prevent the spread of the virus among animals	
**Control** (reduction of the rate of effective contacts)	**Safe herd management practices** ♦ Milk from cows that have tested BLV-negative or milk replacer to feed calves; milk from BLV-infected cows should be treated through freezing or heat treatment ♦ Chemical dehorning or cautery ♦ Disposable needles or needles sterilized by boiling between animals ♦ Clean and disinfect ear tattoo implements ♦ Separate gloves for rectal exploration ♦ Separate calving paddocks for BLV-infected and uninfected cattle ♦ Removal of calves from cows within 24 h of birth but after intake of colostrum ♦ Fly control program	**All herds**
**Prevention** (avoiding introduction)	**Biosecurity measures** ♦ Introduction of animals from certified BLV-infection-free herds ♦ Avoid any contact with infected animals, such as sharing common pastures ♦ Iatrogenic introduction should be prevented	**All herds**
**Surveillance** (maintaining disease-/infection free-status)	♦ Serological surveillance at the herd level involves regular testing of individual or pooled milk or serum samples to detect BLV antibodies ♦ At the regional or country level, surveillance for tumors is conducted during post-mortem inspection of slaughtered animals ♦ Regular testing of representative samples of herds for BLV antibodies from bulk milk samples or individual milk or serum samples	**Free herds/territories**

## Data Availability

No new data were created or analyzed in this study.

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
