# Peer review of "The Global Epidemiology of Bovine Leukemia Virus: Current Trends and Future Implications"

_animals, 2024, doi:10.3390/ani14020297_

Round 1

Reviewer 1 Report

Comments and Suggestions for Authors

Dear authors, thank you very much for this manuscript. Actually, in my opinion, this paper is quite rough and lacking some items, although a review for EBL is an urgent need. I would like to submit to your attention some suggestions, as following.

MAJOR REVISIONS

1. EBL epidemiological situation

Both in the introduction (lines 41-43) and in the paragraph 3/table 1, in my opinion, the EBL epi situation is not in line with the latest updates; also, the sources of data, in my opinion, are not relevant because quite old. Particularly, for the European countries, authors should refer to Regulation UE 2021/620, Annex IV, part I, where Member States or zones thereof with disease-free status from EBL are listed. As you can see, some countries, reported as countries with a certain/unknow prevalence, are actually free, whereas the list of free EBL countries is not complete. Regarding the global situation, the most relevant source of data should be WAHIS, the WOAH database for epi analysis, and not old references from the 90years.

2. diagnostics toll and sequencing analysis

According to international standard/guidelines, analytical sensitivity of serological tests has to be verified  through the international reference serum assay E05. In the manuscript, there is not any mention for this in the paragraph 6, but in my opinion this is an important focus point to be mentioned, for guaranteeing an adequate standardization and validation of the method. Therefore, although several PCR primers are available since many years, a reference exists for labs involved into EBL molecular diagnosis (WOAH terrestrial manual), where some PCR protocols are described as examples. Finally, sequencing analysis is only reported as tool for depth-researches on virus transcriptome (paragraph 11) but a paragraph/part regarding the genetic characterization of BLV and the current distribution of viral genotypes is missing, in my opinion.

3. EBL treatment

According to the current EU legislation, EBL treatment is not allowed, because EBL is listed as a disease to be eradicated. So, even if in the paragraph 9 the authors mention an effective approach for treating EBL, they should clarify that treatment is not an practicable option, in my opinion.

4. prevention/control measures for EBL

In the paragraph 8, control measures of the disease are reported but, in my opinion, the distinction beetween measures required by the law and measures appointed in the farm as good practices, is not clearly characterized. For instance, according to EU legislation, some remarkable conditions are required for bovine movements related to EBL status, and avoiding the introduction of cows from BLV prevalent area (line 403) is not an option; at the same time, using disposable instruments for medical activities (lines 404 ecc) sholud be considered as a routinary practice in a farm where biosecurity measures are effectively implemented.

5. transmission pathways of BLV

In the paragraph 4, the screening breeding bulls is considered as “crucial” to avoid the vertical transmission of BLV; actually, in the EU legislation, it is a required condition for access to a center for semen production. So, in my opinion, authors should report that a relevant legislation is in force, at least in EU.

MINOR REVISIONS

·         LINE 39: “inroduction”, it would be “introduction”.

·         LINE 48: “fever”, maybe it would be “lower”.

·         LINE 129: Author’s name Leisering is written two times.

·         LINE 130: “limph nodes IN THE spleen”?

·         Table 1: in line 145, “68” is reported as reference, but in the caption of table “69” is reported

·         LINE 197: “Bovine Leukemia Virus” is not necessary.

·         LINES 315-317: blood mononuclear cells (PBMC), asymptomatic cattle (AL), persisten lymphocytosis (PL) maybe have to be moved at thye beginning of the paragraph.

Reviewer 2 Report

Comments and Suggestions for Authors

This purports to be a review of the global epidemiology of BLV infection; it tends to focus heavily on the virology part and little on the epidemiological features - in fact, this could be justifiable a manuscript more of a review of the current state of knowledge of the virological aspects.

Section 3 discusses the prevalence in various countries, and is probably not completely inclusive.

Section 7 discusses some of the hazards/ clinical manifestations and section 8 prevention and control but discusses control programs in not much detail (it doesn't mention that infected animals are largely individuals and the key is to find them and eliminate them efficiently - pooling of samples might, for instance, be a cheaper way of identifying them).

The mention of possible human involvement in BLV infection and disease seems speculative and should be deleted from the Abstract/intro - esp since Australia and New Zealand have largely controlled BLV infections in their cattle populations.

In summary, I suggest the manuscript addresses the virology of BLV well, but that is not the topic. Authors should expand the epidemiology, prevalence, diagnostics and control programs part.

Author Response

Dear Reviewer:Thank you for your comments and providing good suggestions for our manuscript. We have revised our manuscript according to your and reviewer's suggestion. My explanation to the comments point- by- point is as follow. The revised manuscript was marked with red color.

  1. The EBL epidemiological situation has been obtained from the World Organisation for Animal Health (WOAH) database, World Animal Health Information System (WAHIS), and the original Figure 1 has been replaced with the diagram obtained.

    The relevant BLV-free countries in Table 1 have also been updated. The data is sourced from Commission Implementing Regulation (EU) 2021/620 of 15 April 2021.UE 2021/620, Annex IV, part I. Prevalence statistics on BLV in various countries and regions have been recently updated through extensive research on Google Scholar and PubMed. The updated data includes information from Uruguay, Ecuador, Mexico, Canada, Brazil, and Russia, reflecting more recent trends beyond the 1990s. Unfortunately, the latest information on BLV prevalence in Saudi Arabia is currently unavailable.

  2. You are correct in emphasizing the importance of targeting and eliminating infected individuals in the control plan for EBL. However, it is important to note that there are currently several challenges due to varying levels of importance attached to EBL in different countries and regions. For instance, EU regulations mandate purification measures, whereas in China, EBL has been reclassified as a Category 3 disease from its previous Category 2 status. Consequently, this divergence in classification has resulted in differences in the prevention and control policies adopted by different countries when dealing with EBL. Therefore, this paragraph aims to provide reference opinions based on EU regulations and research findings, with the hope that the experiences and approaches of different countries can serve as valuable references for the prevention and control of EBL.
  3. There is ongoing controversy regarding the potential of BLV to cause infection in humans. While EBL has been successfully eradicated in New Zealand and Australia, there is still some presence of BLV infection in beef cattle, albeit at a very low rate. Furthermore, a literature report from the 1970s revealed that two out of six chimpanzees, which were fed unpasteurized milk from BLV-infected cows, developed fatal erythroleukemia. Based on your suggestion, we have carefully considered the deletion of the mentioned part from the abstract/introduction of the current manuscript. However, we believe that retaining this section is important for the integrity and comprehensiveness of our study. While we acknowledge that it may potentially spark controversy, we are open to making changes if necessary.
  4. We have made modifications and additions to the sections on BLV prevalence, genotype distribution, detection methods, and treatment. These revisions are highlighted in red in the revised paper. 

In addition to our point-by-point replies to the reviewers’ comments above, we also did our best to meet the standards of required editorial corrections and have made all changes easily identifiable. We hope that our revised manuscript meets your requirements. If any further action is needed, please let us know immediately. We look forward to hearing back from you.

Thank you and best regards.

Yours sincerely,

Dr.Wang

Round 2

Reviewer 1 Report

Comments and Suggestions for Authors

Dear authors,

thank you very much for your intensive work.

In my opinion, all the apported modifications and corrections are adequate,

so no further action is required from my side.

regards.

Author Response

Dear reviewers, thank you again for your careful review and constructive suggestions regarding our manuscript. 

We also wish you all the best in 2024.

Best regards.

Yours sincerely,

Dr.Wang 

Reviewer 2 Report

Comments and Suggestions for Authors

Still not convinced about the case you are trying to make for BLV being a serious public health concern - at least one reference in those cited 22-28 says no, 25 and 28 seem the same, a double-up.

The paragraphs about vaccines and potential drug treatments seem largely speculative and yet the bits about what is known to work - test, cull, and biosecurity management - admittedly hard work - is underdone.

Author Response

Dear reviewer: Thank you for your kind review again.  We have revised our manuscript according to your suggestion. My explanation to the comments point- by- point is as follow. The revised manuscript was marked with  yellow highlights.

1.The repeated references were an error with Endnote. And what we originally meant to convey was that there is still controversy regarding whether BLV is a public health hazard or if people should pay more attention to BLV because BLV DNA can be detected in human breast cancer tissue. This is supported by the available literature, including a recent conclusion on Bovine leukemia virus detection in humans: A systematic review and meta-analysis(Published online 2023 Aug 10. doi: 10.1016/j.virusres.2023.199186. The review mentioned, 'We observed high heterogeneity in the data of frequencies of BLV detection, estimating an overall frequency of 27% (ranging between 17 and 37%).’ While the description in this manuscript may be misleading, we fully agree with your viewpoint and are open to its deletion if deemed necessary.

2.The vaccine section is based on the Enzootic bovine leukosis EFSA Panel on Animal Health and Welfare (AHAW), specifically section 3.5.2 on vaccination. The drugs section is also based on a review of the literature.  Additionally, our research group is currently conducting studies on drugs aimed at reducing BLV PVL and has made some initial findings, although these have not yet been published. If there are any aspects that require modification, please indicate them. We have provided a detailed description of the test, cull, and biosecurity management section in paragraph 8, and have included Table 4, which outlines the control tools for EBL. Kindly review these additions.

We hope that our revised manuscript meets your requirements. We are continuously reviewing the manuscript to prevent issues such as misspelled words and duplicate references. We appreciate your understanding regarding these minor errors.. We look forward to hearing back from you.

Thank you and best regards.

Yours sincerely,

Dr.Wang